# Nanoliposomes and Tocosomes as Multifunctional Nanocarriers for the Encapsulation of Nutraceutical and Dietary Molecules

**DOI:** 10.3390/molecules25030638

**Published:** 2020-02-01

**Authors:** Ali Zarrabi, Mandana Alipoor Amro Abadi, Sepideh Khorasani, M.-Reza Mohammadabadi, Aniseh Jamshidi, Sarabanou Torkaman, Elham Taghavi, M.R. Mozafari, Babak Rasti

**Affiliations:** 1Sabanci University Nanotechnology Research and Application Center (SUNUM), Tuzla 34956, Istanbul, Turkey; alizarrabi@sabanciuniv.edu; 2Dept. of Food Hygiene, The Faculty of Veterinary Medicine, Shahrekord City Branch, IAU University, Shahrekord 8813733395, Iran; alipour.mandana9@gmail.com; 3Department Food Science and Industries, Faculty Agriculture, Shahid Bahonar University of Kerman, 22 Bahman Blvd., Kerman 7616914111, Iran; khorasany@uk.ac.ir; 4Department of Animal Science, Faculty of Agriculture, Shahid Bahonar University of Kerman, 22 Bahman Blvd., Kerman 7616914111, Iran; mrm@uk.ac.ir; 5Australasian Nanoscience and Nanotechnology Initiative, 8054 Monash University LPO, Clayton, VIC 3168, Australia; a.jamshidi1382@gmail.com (A.J.); Sarabanouu2560@gmail.com (S.T.); elham.taghavi@umt.edu.my (E.T.); 6Faculty of Fisheries and Food Science, Universiti Malaysia Terengganu, 21030 Kuala Nerus, Terengganu 21030, Malaysia; 7Faculty of Food Science & Nutrition, Universiti Malaysia Sabah, Jalan UMS, 88400 Kota Kinabalu, Sabah 88400, Malaysia

**Keywords:** encapsulation, food technology, Mozafari method, nutraceuticals, nanoliposome, supplements, tocosome

## Abstract

Nanoscale lipid bilayers, or nanoliposomes, are generally spherical vesicles formed by the dispersion of phospholipid molecules in a water-based medium by energy input. The other nanoscale object discussed in this entry, i.e., tocosome, is a recently introduced bioactive carrier made mainly from tocopheryl phosphates. Due to their bi-compartmental structure, which consists of lipidic and aqueous compartments, these nanocarriers are capable of carrying hydrophilic and hydrophobic material separately or simultaneously. Nanoliposomes and tocosomes are able to provide protection and release of sensitive food-grade bioactive materials in a sustained manner. They are being utilized for the encapsulation of different types of bioactive materials (such as drugs, vaccines, antimicrobials, antioxidants, minerals and preservatives), for the enrichment and fortification of different food and nutraceutical formulations and manufacturing of functional products. However, a number of issues unique to the nutraceutical and food industry must first be resolved before these applications can completely become a reality. Considering the potentials and promises of these colloidal carrier systems, the present article reviews various aspects of nanoliposomes, in comparison with tocosomes, including the ingredients used in their manufacture, formation mechanisms and issues pertaining to their application in the formulation of health promoting dietary supplements and functional food products.

## 1. Introduction

Encapsulation and/or entrapment of bioactive compounds is a process in which small solid particles, or droplets of liquids or gases, are separated from other particles and from the external medium using a thin film or a vesicular system [1,2]. This process can mainly be physical (e.g., encapsulation or entrapment by colloidal carriers), chemical (e.g., by chemical conjugation of active material to the carrier molecules) or electrostatic adsorption to the surface of metal particles [1,2]. A wide range of coating materials can be employed in the field of food and nutraceuticals. These include celluloses, gums, lipids, phospholipids and proteins [3,4]. Depending on the carrier material and the process method used, encapsulation may provide several advantages to the encapsulated compounds and the finished product as well. Some of these advantages and benefits are listed in Figure 1.

In the nutraceutical, dairy and food industries, encapsulation technology is employed to stabilize sensitive components, control the release of core material and to physically separate reactive or incompatible ingredients and thereby increasing product shelf life [5,6]. Encapsulation techniques can be applied to protect the structure and function of food compounds and nutraceutical material and to improve their bioavailability. Nanovesicles and nanocarriers offer the food processor several advantages by ensuring against nutritional loss, incorporating time-release or pulsative mechanisms into the formulation, masking or preserving aromas and flavors, and transforming liquids into the more easily handled solid products [5,6].

One of the encapsulation techniques that has captured much attention in the pharmaceutical, food and cosmetic industries is the use of phospholipid vesicles known as liposomes. Liposomes are scientifically defined as closed, continuous, bilayered structures made mainly of phospholipid molecules [7]. The nano-scale forms of liposomes are called nanoliposomes [8]. These encapsulation techniques, along with a recently introduced bioactive carrier system called “tocosome” [9], are the subject of the present review article. Their physicochemical attributes, mechanism of formation, ingredient molecules, along with their potential applications in the area of food and nutraceutical science and technology are also reviewed. 

## 2. Nanoliposome Technology

Nanoliposome is a spherical or oval vesicle consisting of a phospholipid bilayer (or two or more bilayers separated by aqueous interphases) entrapping a central liquid core [10,11]. During the vesicle formation process, the hydrophilic material becomes entrapped or encapsulated within the aqueous regions (including the central core), while the hydrophobic molecules are incorporated in the bilayer membrane(s) or the lipidic domains of the vesicles. Release of the entrapped compounds can be either a gradual process resulting from diffusion through the bilayers, or almost instantaneous as a result of vesicle disruption caused by changes in pH, osmotic pressure, ionic strength or temperature. The phospholipid vesicles are ideal systems for the entrapment of different types of therapeutic and diagnostic agents (theranostics), vaccines, nucleic acid drugs and minerals (e.g., Ca^2+^, Mg^2+^) separately or in combination [12,13,14]. Nanoliposomes are also being considered for gene therapy applications as well as treatment of metabolic disorders [15,16]. In addition, there are a number of potential applications for nanoliposomes in the food, feed and nutraceutical sectors as well. They are able to protect sensitive material from degradation, allow incompatible ingredients to be encapsulated together, confine unpleasant odors and prevent unpleasant flavors from interacting with taste buds. Increased efficacy of the encapsulated food additives may allow significant reductions in the required amounts of these additives. The controlled release property of nanoliposomes may allow compounds such as enzymes or antimicrobial agents to be added much earlier than their action is required, without adverse effects. Applications of lipid vesicles, including nanoliposomes, in the pharmaceutical and cosmetics industries are widespread. However, issues of food safety and cost-effectiveness have limited their applications in the food systems. Nevertheless, the application of specialized techniques for large-scale manufacture of the lipid-based vesicles, and commercially available lecithin ingredients of relatively modest cost, may solve these problems [17,18].

There is a range of synthetic and natural phospholipid ingredients available for the formulation of lipid vesicles. The most common phospholipid molecule, which is employed in nanoliposome preparation, is phosphatidylcholine (PC). The cylindrical-shaped PC molecule aggregates into bilayer planar sheets, reducing the thermodynamically unfavorable interactions between the bulk aqueous phase and the hydrocarbon chains (hence reducing the level of energy) [17]. This energy-minimization results in any hydrophobic material in the aqueous media being internalized by the bilayer membrane. Nevertheless, the bilayer arrangement still has exposed hydrophobic fatty acids at the edges of the planar sheet. To completely eliminate the unfavorable contact between these fatty acids and the water molecules, the sheet folds to bring the edges together to form a closed vesicle, hence enclosing a portion of the aqueous phase, which becomes the central core of the vesicle. Molecules dissolved in the aqueous phases may become trapped within the carrier system during this process. Consequently, nanoliposomes and tocosomes have the capacity to encapsulate both hydrophilic and hydrophobic materials within a single structure. Although the vesicular arrangement is at the minimum thermodynamic energy level [19], for vesicle formation to occur the system has first to be supplied with a minimum quantity of energy called “the activation energy”. This required energy input could be either physical, mechanical, thermal, acoustic (e.g., ultrasonication), or a combination of these [20].

## 3. Tocosomes

Tocosome is a colloidal and vesicular bioactive carrier system, the main constituents of which are phosphate-group-bearing alpha tocopherols [9]. However, like nanoliposomes, they can also accommodate sterols, proteins and polymers in their structure. The phosphorylated form of alpha-tocopherol, known as alpha-tocopherol phosphate (TP), is present naturally in human and some animal tissues as well as in certain food compounds [21,22]. It has recently been reported that the TP molecule is naturally present in certain fruits, green vegetables, cereals, dairy products, as well as in different nuts and seeds [23]. TP is composed of a phosphate group attached to one hydrophobic chain (phytyl tail) made of three isoprene units. Di-alpha-tocopherol phosphate (T_2_P), a closely related molecule to TP, is composed of two phytyl chains. However, unlike phosphatidylcholine and some other phospholipids, the hydrophobic phytyl chains of T_2_P cannot align in parallel position due to the presence of bulky isoprene side-chains (Figure 2). Consequently, the geometric shape of T_2_P molecule is conical, while the TP molecule is cylindrical-shaped (similar to PC molecule). 

Clinical investigations have shown that TP and T_2_P molecules possess many health benefits such as anti-inflammatory, atherosclerotic-preventing and cardioprotective properties [24,25]. In addition, the inhibitory effect of TP molecule towards tumor invasion has also been reported [26]. Studies also showed that α-TP protected primary cortical neuronal cells from glutamate-induced cytotoxicity in vitro and reduced the levels of lipid peroxidation products in the plasma and liver of mice in vivo [27]. Tocosomal formulations containing different phospholipid molecules (in addition to TP and T_2_P components), and varying combinations of cholesterol, have been employed recently for the entrapment and controlled-release of the anticancer drug 5-fluorouracil [9]. Being composed of molecules with exquisite health benefits and strong antioxidant activities, tocosomes have great potential for applications in the formulation of food and nutraceutical products, as explained in the following sections.

## 4. Differences between Tocosomes and Nanoliposomes

The main underlying difference between a tocosome and a nanoliposome is the distinction between their main ingredients TP/T_2_P and phospholipids. Alpha-tocopherol phosphate (also called α-tocopheryl phosphate or tocopherol phosphate ester) is in fact a phosphoric acid ester of α-tocopherol (vitamin E), possessing the hydroxyl group of tocopherol [28]. The phosphate ester of alpha-tocopherol is present in certain animal and plant tissues [23]. The TP molecule can exist in either the natural form (i.e., RRR, d) or the synthetic form (i.e., all-racemic, dl), as is the case with the vitamin E molecule. One of the stereoisomers in all-racemic alpha-tocopherol molecule is 2R, 4′R, 8′R (designated as RRR) that is the only stereoisomer found in nature. Another tocopherol derivative has also been identified as the bis-tocopherol phosphate ester, or di-alpha-tocopherol phosphate (i.e., T_2_P, Figure 2). The T_2_P molecule can be synthesized through esterification of two tocopherol molecules and one phosphate molecule. A number of other synthetic derivatives of vitamin E have also been synthesized which have novel or tocopherol-related biological activities. These synthetic molecules can be converted by the esterase enzymes to the natural form of vitamin E [9,24].

Over recent years, explicit effects for each individual vitamin E-analogue have been defined at cellular level. These effects are resulted from either modulating signal transduction and/or gene expression. They possibly reflect particular interactions of each of the vitamin E-analogues with lipids, structural proteins, enzymes and/or transcription factors. There are ambiguities with respect of classification of tocopheryl-phosphate molecules and in some cases, they are erroneously referred to as phospholipids. This is while phospholipids are similar to triglycerides except that the first hydroxyl moiety of the glycerol molecule has a polar phosphate group instead of the fatty acid chain. Phospholipids are amphipathic (amphiphilic), being both hydrophilic and hydrophobic. The head group of a phospholipid molecule is hydrophilic and its fatty acids (acyl chains) are hydrophobic. The phosphate moiety of the phospholipid head group is anionic (possessing negative zeta potential). Phospholipids are a class of lipids, which are a major ingredient of all bio-membranes. Each phospholipid molecule contains a diglyceride, a phosphate group, and in addition a simple organic molecule such as a choline moiety. A phospholipid molecule can also be defined as a lipid, which in its simplest form is composed of a glycerol bonded to two fatty acid chains and a phosphate-bearing group [24,25,26].

It is noticeable that while definition of a phospholipid molecule necessitates presence of a phosphate group, two fatty acid tails and a glycerol linker, the tocopherol phosphates are composed of a chroman head (with two rings: one phenolic and the other heterocyclic) and a phytyl tail with 3 isoprene side-chains. Moreover, tocopherols and their derivatives (i.e., TP and T_2_P molecules) do not contain glycerol group, that is an indispensible chemical part of phospholipids [29]. Chemical structure of a tocopheryl phosphate molecule, in comparison with a phospholipid molecule, is depicted in Figure 3. Although differences in the chemical constituents of nanoliposomes and tocosomes are undeniable, these carrier systems behave pretty much the same with respect of their drug delivery mechanisms and release behavior. They are both bilayer colloidal systems composed of amphiphilic molecules and as such can be utilized for the encapsulation, entrapment and controlled release of different types of bioactive compounds. Consequently, both of these carrier systems can be used successfully in the food, feed and nutraceutical industries for the same applications keeping in mind that the ingredients of tocosomes possess more health benefits compared to phospholipids [9,11].

## 5. Mechanism of Formation of Nanoliposomes and Tocosomes

Nanoliposomes and tocosomes are formed through the assembly of amphipathic molecules, mainly but not exclusively phospholipids (in the case of nanoliposome) and tocopheryl phosphates (in the case of tocosome), in an aqueous environment, as a result of hydrophilic/hydrophobic interactions and van der Waals forces [30,31]. In addition to lipids, phospholipids, TP and T_2_P molecules, nanoliposomes and tocosomes may contain other molecules including proteins and carbohydrates in their structure to increase their stability or as a targeting strategy [32,33]. When the amphipathic ingredients are churned in an aqueous medium, the shape of the molecules is a major factor in determining which of a variety of different structures is most likely to be produced. The most abundant phases found in the amphiphilic systems of interest are the micellar solutions with regular (L_1_) or reversed (L_2_) aggregate structures, lamellar (Lα) liquid-crystalline phases, normal (H_1_) or reversed (H_2_) hexagonal liquid-crystalline phases and a number of different cubic liquid-crystalline phases [34], some of which are depicted in Figure 4.

When churned in aqueous media under certain conditions of temperature, pH, pressure and agitation, the amphipathic molecules arrange themselves as spherical bilayer structures via hydrophilic/hydrophobic and Vander Waals interactions and form tocosomes or nanoliposomes. As a consequence, the hydrophilic groups of the ingredient molecules face the interior and exterior aqueous phases of the vesicles while the hydrophobic fragment of each of the monolayers face each other in an attempt to avoid contact with water molecules. It should seriously be considered that, although generally suggested, formation of drug delivery vesicles, such as tocosomes and lipid bilayers, is not a spontaneous process. An adequate quantity of energy, in the right form, must be provided to the system for these carrier systems to form [20]. The underlying biophysical principles and the mechanism of formation of bioactive carrier systems, including tocosomes and nanoliposomes, are described in detail by Lasic [35,36,37] as well as Mozafari et al. [8,20]. In brief, bilayered colloidal carriers are formed when their ingredient molecules are placed in aqueous media, such as distilled water, buffer or an isotonic solution, and afterwards form bilayer vesicles once sufficient level of energy is supplied. This is a strictly required stage in order to overcome an energy barrier for curving the planar lipid/phospholipid/tocopheryl phosphate bilayers and form spherical vesicles. Input of energy (e.g., through homogenization, agitation, heating, sonication, microfluidization, etc.) results in the arrangement of the amphipathic molecules, in the form of bilayer structures, to achieve a thermodynamic equilibrium in the aqueous media [8,20,35,36,37]. Based on these principles, several methods have been invented and introduced for the manufacture of lipid vesicles and tocosomes, some of which are listed in Table 1. These methods are generally classified as low-energy and high-energy techniques. Low-energy consuming procedures include solvent injection, solvent diffusion and Mozafari method. High-energy consuming techniques include microfluidization, high-pressure homogenization (hot and cold), and the sonication methods [38,39].

## 6. Applications in the Food Industry

There has been less development of nanoliposome and tocosome technology in the food and diet industries compared with that seen in the pharmaceutical and cosmetics industries. Despite this, it is believed that the agrifood industry has the largest number of potential applications for nanobiotechnology [40]. The limited development to date has not been due to a lack of potential applications, but to difficulties in finding safe, low-cost ingredients and economical processing methods suitable for producing large volumes of nanoliposomes/tocosomes with batch-to-batch consistency. The recent development of Mozafari method [41,42] and pro-liposome techniques [43] offers possible solutions to many of the food processing problems. Moreover, ongoing research into the application of cost-effective commercial lecithin ingredients may lead to suitably low product costs [44].

The application of tocosomes and nanoliposomes as potential carriers to encapsulate and deliver food ingredients and nutraceutical compounds is relatively an innovative technology. Studies thus far indicate the potential of carrier systems for improving the flavor of ripened cheese using accelerated methods, the targeted delivery of functional food ingredients, the synergistic delivery of tocopherols and ascorbic acid for enhancing antioxidant activities in foods, and the stabilization of minerals, such as calcium and iron, in milk and other drinks [45]. In the food, diet and nutraceutical industries, nanoliposomes and other lipidic carriers have been employed to encapsulate flavoring and nutritive agents. They have also been suitable candidates to entrap and deliver antimicrobial preservatives in order to improve product shelf life. Some of the potential applications of nanoliposomes and tocosomes within the food industry are discussed in the following sections.

### 6.1. Applications in Dairy Products

There are several potential applications for the encapsulation technologies in the dairy industry, which range from the protection of sensitive molecules and compounds to increasing the efficacy of food additives. A number of scientific literature claim that it is possible to modify the pharmacokinetic characteristics of drugs, herbs, vitamins and even enzymes through employment of nanocarrier systems (e.g., see [46,47]). Furthermore, some lipid vesicles have been employed for targeted delivery of the encapsulated compounds in dairy products [48]. The main applications thus far have been aimed to change the texture of food components, accelerate cheese ripening, develop new tastes and sensations, regulate the release of flavors and increase the absorption and bioavailability of nutritional compounds. In the dairy industry, the first application of the lipid-based vesicles was in the cheese ripening process [49]. In addition, different types of lipidic carriers such as nanoliposomes have been utilized for the entrapment or encapsulation of antimicrobials, enzymes and minerals (e.g., calcium, iron) in different dairy products [11].

The ability of tocosomes and nanoliposomes to provide targeted delivery of the encapsulated material in specific areas of the food system is highly beneficial for the dairy industry. For example, the employment of proteinase enzymes encapsulated in the lipid vesicles can significantly reduce the time and cost of cheese ripening [10,18]. Cheese ripening process comprises a series of complex chemical reactions including glycolysis, lipolysis and proteolysis that result in desired aroma, taste and texture characteristics in different types of cheese. The most important stage of cheese ripening process is proteolysis that changes the aroma and taste of the dairy product. Since the cheese ripening process is slow and costly, and sufficient time for the growth and activity of spoilage organisms is available, there are attempts to shorten this period by adding the flavoring agents, enzymes, texture improving components and to protect cheese against spoilage through incorporation of preservatives using encapsulation technology [45]. To minimize the maturation time and avoid bitterness and texture defects, enzyme release should be controllable and progressive. By encapsulating the enzymes in nanoscale carrier systems, the enzyme will be released in a prolonged manner from the vesicles. Studies have shown that nanoliposome-encapsulated enzymes mainly concentrate in the curd during the formation of cheese. This is while free (unencapsulated) enzymes are typically distributed in the whole-milk mixture evenly. As a result, this leads to a very low retention of the flavor-producing enzymes in the curd. It was also detected that the lipid vesicles were degraded naturally in the cheese matrix after whey separation. Consequently, this allowed full contact between the protein matrix and the cheese ripening enzymes [50,51].

One of the most employed enzymes in cheese ripening is proteinase, although lipase and flavourzyme are also being employed (encapsulated or in their free form) to some extent. Accelerated Cheddar cheese ripening, without producing off flavors or causing texture defects, has been achieved by using encapsulated bacterial or fungal proteinase [52]. Enzymes show more stability in high concentration mediums. Furthermore, it is possible to utilize stabilizing components such as thermostabilizers (e.g., sugars), encapsulated together with the enzyme to protect it against high temperatures during food processing steps [53,54]. By fine-tuning the composition of nanoliposomes and tocosomes, the time, duration and the rate at which the enzyme is released can be optimized. The gradual release of the encapsulated material in the cheese matrix avoids the bitter taste formation caused by free enzymes. Addition of unencapsulated enzymes to the milk has a substantial adverse influence on the chemical properties of cheese curd and decreases the efficiency of cheese manufacturing due to rapid hydrolysis of the casein. The effectiveness of nanoliposomal protease on the properties of cheese curd depends on the type of enzyme, its entrapment efficiency and the concentration of the vesicles in the curd. Use of encapsulated protease inhibited decrease in the pH value and also prevented calcium release in the whey of Saint Paulin cheese [55]. Consequently, it caused an improved taste and texture as well as acceleration in cheese ripening when compared with cheese production using the free (unencapsulated) enzyme (for a review see: [56]).

Another enzyme employed in the dairy industry is flavourzyme, which is a peptide produced by Aspergillus oryzae and is extensively utilized for the hydrolysis of different proteins [57]. In a study conducted by Jahadi and co-workers [58], the heating method (Table 1) was employed to incorporate flavourzyme into nanoliposomes. The enzyme-loaded vesicles were mixed with cow milk, which resulted in the formation of white-brined cheese. The product was evaluated with respect of proteolysis degree, chemical constituents of whey plus curd, and yield. An entrapment efficiency of approximately 25% was achieved for the flavourzyme enzyme and the nanoliposomal formulation caused no apparent changes in the sensory characteristics of the cheese. Furthermore, the chemical composition of both curd and whey remained constant when compared to the control sample statistically [58]. In another study, flavourzyme-loaded nanoliposomes were added to white cheese, and the impact of different factors (i.e., concentration of the loaded enzyme, brining time, and ripening period) on the sensory properties and the proteolytic activity was evaluated. All factors significantly influenced proteolysis, although ripening period was the most influential parameter. The study revealed that cheese samples produced under the following conditions were the best products with respect of proteolysis index and sensory attributes: (i) loaded enzyme content of 0.3% w/w, (ii) brining time of 8 h, and (iii) ripening time of 1 month [59].

In addition to enzyme encapsulation, tocosomes and nanoliposomes can be employed for the fortification of different products with health-benefit agents. As a result of the increased demand for products with high nutritional value, the food industry in general, and the dairy industry in particular, seek to fortify their products with minerals, vitamins and other functional ingredients. A few studies have reported fortification of food products with encapsulated vitamins employing lipid vesicles [60,61]. Of particular interest is vitamin D, which plays a very important role in the structure and function of cartilage and bone tissue. Fortification of cheese with this vitamin can be achieved through one of the following procedures: i) using a derivative of vitamin D dissolved in water, ii) dissolving the crystalline form of vitamin D in the milk cream and adding the fortified cream to milk, or iii) utilization the encapsulated form of vitamin D in a suitable carrier system [56]. Vitamin D retention during the storage of cheese products supplemented via the above-mentioned three methods was 42.7%, 40.5% and 61.5%, respectively. It is apparent that encapsulation of vitamin D provides the highest efficiency. The stability of vitamin D during the manufacture and ripening of cheese over a 7-months period was highest by employing lipid vesicles for the encapsulation and controlled release of vitamin D [62]. Co-encapsulation of vitamin C (into the aqueous phase) and vitamin E (in the lipid phase) of the bilayered lipid vesicles has also been reported. This is an example of a potential strategy to incorporate two vitamins into food systems simultaneously [11,63,64].

### 6.2. Encapsulation of Minerals

A deficiency of certain essential minerals can adversely affect human health. However, it is not usually possible to add them directly into food systems since fortification of foods by some minerals (e.g., calcium, magnesium, iron) has some draw backs such as complexation with other food ingredients, causing changes in odor, taste and appearance, and increasing the risk of oxidation [65,66]. Today, iron deficiency is highly recognized as one of the major nutritional deficiencies worldwide. This condition is mainly due to insufficiency in dietary intake of iron, lack of its bioavailability, or both [67]. Iron shortage in blood should not be ignored as it may cause anemia, when the blood hemoglobin level falls below standard levels [68]. Since there is an antagonism between iron and milk’s calcium content, milk is low in iron and it can cause iron deficiency in young children [69]. Hence, it is necessary to enrich the milk with iron and to protect it against iron-calcium antagonism, fat oxidation consequences and metallic off-flavor. To achieve this, supplementation of milk with iron encapsulated in safe and non-toxic carriers has been proposed. Towards this end, highly soluble form of iron (i.e., ferrous sulfate) is preferred due to its cost effectiveness and high bioavailability [70]. Generally, iron salts are classified into three groups based on their solubility: (i) the water-soluble, (ii) the acid-soluble, and (iii) the poorly soluble salts. The water-soluble iron salts (e.g., ferrous sulfate and ferric chloride) are characterized by the highest bioavailability and low cost. Studies showed that the water-soluble ferrous salts possess higher bioavailability than the ferric salts [71]. However, water-soluble iron salts induce undesirable changes in the fortified food products, particularly in enhancing lipid oxidation, as well as odor and color changes. Iron sulphate causes rapid color and flavor changes when added directly to food products. However, yoghurt containing encapsulated iron sulphate showed no color change after 2 weeks [72].

Recently, Simiqueli and colleagues [73] reported that fortification of food products with FeSO_4_ is an effective approach for preventing iron deficiency (anemia). However, direct addition of this mineral to foods causes unfavorable sensory qualities including the darkening of the matrix and unpleasant off-flavors. Encapsulation of FeSO_4_, or other salts such as MgCl_2_, in the internal aqueous phase of W/O/W double emulsions or nanoliposomes may effectively prevent these adverse changes [74]. In the iron-loaded nanoliposome formulations, using antioxidants such as vitamin E or ascorbic acid in the structure of nanoliposomes is a common practice in order to protect the ferrous ion against oxidation [75]. When using tocosomes, this requirement is alleviated because of the strong antioxidant activity of tocosomal ingredients TP and T_2_P [22,25].

Another example of a mineral considered for food fortification is magnesium. It is an essential mineral compound, which is associated with lowering the risk of some clinical disorders including cardiovascular disease, hypertension, type 2 diabetes and muscular weakness. Consequently, fortification of food systems by magnesium can result in the improved human health [65]. Bonnet and colleagues [76] formulated W/O/W emulsions using different lipid phases (rapeseed oil, olein, olive oil, and miglyol) to encapsulate magnesium in the inner aqueous droplets. They used polyglycerol polyricinoleate (PGPR, E476) as a lipophilic and sodium caseinate as a hydrophilic emulsifier. The nature of oil employed in the formulation affected magnesium release from the emulsions so that the higher delivery rates were observed for the oils with a lower viscosity and containing the higher percentage of saturated fatty acids. Magnesium release was related to diffusion mechanisms in a characteristic rate that changed by time. In addition, various thermal treatments (pasteurization processes) could not affect the stability of the W/O/W emulsions. It is reported that the heat treatment of milk with added free magnesium caused protein coagulation within a few minutes, whereas encapsulation of magnesium prior to addition to milk resulted in no adverse effect at all [76].

### 6.3. Encapsulation of Antioxidants

Oxidation reactions are the main cause of deterioration of oils, fats and lipid-based nutrients and, consequently, result in decreased nutritional value and sensory qualities of food products. It has been suggested that the oxidation of biomolecules is involved in several chronic and age-related disease, including cancer, cataract, cardiovascular disorders, rheumatoid arthritis and diabetes mellitus [77]. A possible approach to provide protection against oxidation is employment of compounds, which possess antioxidant properties. Antioxidant molecules or compounds can be defined as substances that significantly inhibit or delay oxidation of a substrate while present in small concentrations [78]. Antioxidants added to food products, on the other hand, may be defined as any substance capable of delaying, retarding or preventing the development of rancidity or other deteriorations due to oxidation [79]. Before being accepted for supplementation into food products, antioxidants must meet several criteria. They should maintain their activity and protect the finished product even on long-term storage; they should not impart a foreign color, odor or flavor to the food; they should be stable during heat processing; they should be easy to incorporate and be effective at low concentrations [63]. A possible approach to meet all these criteria is the employment of encapsulation technologies.

The entrapment (within the amphiphilic bilayer) or encapsulation (within the aqueous core) of antioxidants is a typical example of the potential benefits of nanocarrier technologies, including nanoliposomes and tocosomes. The trend towards the replacement of saturated fat with unsaturated fat in the food products has increased the susceptibility of many fat-containing foods to oxidation, especially in the emulsion-based food products such as mayonnaise, spreads and margarines. Until recently, mainly lipophilic antioxidants have been employed to counteract food oxidation. However, the amphiphilic TP and T_2_P molecules have been recently introduced as very potent alternatives to the traditional antioxidants [21,22]. These phosphorylated molecules are much more potent than the tocopherol molecules they are derived from [25,27]. In 2017, Mozafari and co-workers devised a novel drug delivery system using TP and T_2_P molecules called tocosome [9], which by itself (empty vesicle without encapsulated material) is an ideal antioxidant system. Once another antioxidant is encapsulated in tocosomes, they will construct a multicomponent, synergistic antioxidant delivery formulation for applications in different food systems.

Ascorbic acid (vitamin C) and α-tocopherol (vitamin E) can act in a synergistic manner as natural antioxidants. The α-tocopherol molecule reacts with peroxy radicals in the continuous phase of the food to form α-tocopheroxyl radicals, which are less effective than peroxy radicals in the oxidation chain reaction initiation [80]. The α-tocopheroxyl radical can be reduced to α-tocopherol by ascorbic acid. This regeneration extends the antioxidant effect of the α-tocopherol. However, α-tocopherol is hydrophobic and therefore cannot interact with the water-soluble ascorbic acid. It is possible to use lipid-soluble derivatives of ascorbic acid (e.g., ascorbyl palmitate); however, its effective dispersion requires high temperatures, increasing the likelihood of oxidation problems in the food system. An alternative may be to use an encapsulation technology such as nanoliposome or tocosome for co-delivery of antioxidants [9,63].

Different encapsulation technologies including tocosomes and nanoliposomes can encapsulate and release two or even more materials with different solubilities simultaneously. One example is the incorporation of two antioxidant agents, i.e., α-tocopherol (a lipid-soluble molecule) and glutathione (a water-soluble molecule), in the same vesicle [81,82]. Another example of a bifunctional carrier system is a nanoliposomal system encapsulating both ascorbic acid and α-tocopherol [83].

Liposome-entrapped α-tocopherol has been shown to be more effective at preventing oxidation in oil-in-water emulsions than when it is in the free form and dissolved in the oil phase of the emulsion [72]. It is known that oxidation occurs first at the water/oil interface. If the nanoliposome/tocosome is situated at this interface, the α-tocopherol in the membrane could reduce the peroxy radicals before these radicals initiate oxidation (Figure 5). Ascorbic acid entrapped in the aqueous phase of the carrier system could regenerate the α-tocopherol. Encapsulation of the ascorbic acid would minimize the degradation of this molecule by other food components and ensure maximum α-tocopherol regeneration.

The scavenging activity of antioxidants can be assessed using techniques such as oxygen radical absorbance capacity or 2,2-diphenyl-1-picrylhydrazyl radical assays [84]. These techniques are the most widely applied even though they suffer from antioxidant solubility problems and they do not represent naturally occurring media. Other methods, such as conjugated autoxidizable triene, apolar radical initiated conjugated autoxidizable triene, or EPR spectroscopy scavenging assays involve evaluation of antioxidant activity in complex media such as emulsions or microemulsions [85,86,87]. Generally, scavenging capacity in media with multiple polarities tends to resemble the naturally occurring food matrices thus giving relatively more accurate antioxidant activities for food and nutraceutical applications [88].

### 6.4. Encapsulation of Food Preservatives

Natural antimicrobials are attracting more attention mainly due to the negative effects of chemical preservatives on human health as well as significant increase in the number of antibiotic-resistant microorganisms [89]. Bacteriocins and essential oils are common examples of natural antimicrobial agents that can be employed in food systems to extend product shelf life and protect them against microbial spoilage [90]. Essential oils are highly volatile aromatic substances, which can be extracted from various parts of plants and herbs (e.g., bark, stems, leaves, roots, flowers, fruits, and seeds). They not only possess significant inhibition actions on different kinds of pathogens, but also provide several medicinal benefits such as antioxidant, anticancer, antiviral, anti-allergic, and anti-inflammatory properties. Moreover, it is possible to utilize essential oils and their constituents as flavoring agents in different food products, nutraceuticals, cosmetics, and pharmaceutical formulations [91,92]. However, some hurdles, including poor stability, high reactivity, low water solubility, and volatility, make direct employment of essential oils and their components into food systems difficult [93]. They easily undergo degradation during storage and throughout processing procedures of food products, leading to a significant decline in their efficiency. In addition, they are inappropriate for water-based foods because of dispersibility problems and their adverse effects on the sensory characteristics of food products [94,95]. A possible solution to these shortcomings is employment of encapsulation technology and bioactive carrier systems.

Bacteriocins, on the other hand, can be defined as antimicrobial peptides obtained from some bacteria with inhibitory or bactericidal activities towards other types of bacteria [96]. Among the naturally occurring bacteriocins, the antimicrobial nisin has attracted most interest in research and development since it is a generally recognized as safe (GRAS) bacteriocin approved for human use by the regulatory institutes including FDA [97]. Nisin consists of 34 amino acids synthetized by some species of Lactococcus lactis and is categorized into two different types, i.e., nisin A and nisin Z. For food applications, nisin Z is superior to nisin A because of its good dispersibility and stability in food matrices [98]. Nisin potentially contributes to the prevention of growth of a wide variety of Gram-positive strains (e.g., Staphylococcus and Listeria) as well as spores formed by Clostridia and Bacilli [99]. Despite all these advantages, if added to food systems directly, its effectiveness will considerably diminish due to its poor distribution into food systems, degradation by harsh environmental conditions and proteolytic enzymes, and its unfavorable interactions with other food components [41,100]. Bouksaim and co-workers [101] detected that presence of free nisin Z declines the efficiency of cheese starters, leading to low sensory qualities.

Nanoliposomal and tocosomal carrier systems can be employed as appropriate and efficacious strategy to overcome the aforementioned problems. They are able to prevent unfavorable interactions between the natural antimicrobials and food components, enable programmed and sustained release, along with improved stability and consumer satisfaction with respect of food sensory qualities. They are also able to slow down development and spread of antibiotic-resistant bacteria. Because of structural resemblance of the lipid vesicles to cell membranes, liposomal formulations simplify the cellular transport and provide effective interaction of antimicrobial compounds with microorganisms, enhancing antimicrobial action [41,102]. Several research studies have reported the encapsulation of essential oils and nisin using different types of nanocarriers as listed in Table 2.

Da Silva Malheiros and colleagues [103] designed nanoliposomes to encapsulate nisin and BLS P34 (a bacteriocin-like substance produced by Bacillus sp. P34) employing thin-film hydration method using soybean PC (with and without cholesterol) for enhancing the antimicrobial efficiency against Listeria monocytogenes in Minas frescal cheese. The formulated nanoliposomes possessed very high encapsulation efficiencies (i.e., 88.9% and 100% for nisin and BLS P34 respectively) and zeta potential values greater than −30mV, which indicate stable formulations. The encapsulated forms of the antimicrobials were able to decrease L. monocytogenes counts in comparison with the control (unencapsulated) sample when stored at 7 °C for 3 weeks. The presence of cholesterol caused a significant loss in the inhibitory capacity of the encapsulated bacteriocins. Authors reported that the maximum antibacterial activity was related to nanoliposomes composed of PC after storage for 10 days [100].

In another study, Pinilla and Brandelli [104] applied the thin film hydration technique for the nanoliposomal entrapment of nisin individually and together with garlic extract. They evaluated the antibacterial efficiencies of the formulations toward some pathogenic bacterial strains including Escherichia coli, L. monocytogenes, Staphylococcus aureus and Salmonella enteritidis during storage of the whole UHT milk at 37 °C. Nanoliposomal formulations showed desirable physicochemical characteristics and also higher capability to inhibit growth of bacteria compared with the free form of the antimicrobial agents. Authors reported that the encapsulated bacteriocins had a better performance against Gram-positive species [104].

In a study by Cui and team [105,106], soy lecithin was used to construct nanoliposomes with the aim of improving stability and bactericidal efficacy of clove oil. The antimicrobial efficiency of the formulation was assessed towards S. aureus and E. coli in tofu. Results showed that nanoliposomes extended the inhibitory effect of clove oil on S. aureus owing to the pore-forming toxins and therefore rapid release of the antimicrobial compounds. However, such activity was not observed for E. coli. In another research, Peng et al. [107] successfully developed a nanoliposomal eugenol formulation using a mixture of phospholipids, cholesterol and Tween 80 via dynamic high-pressure microfluidization technique. The nanoliposomal formulation exhibited desirable properties and prolonged stability. There were no oil droplets after 2 months of storage at both room and refrigeration temperatures, which confirmed positive effect of nanoliposomes on promoting water solubility of the lipophilic eugenol. Moreover, the phospholipid vesicles provided sustained release properties for eugenol [107].

Incorporation of Zataria multiflora essential oils into nanoliposomes was carried out using soy PC via thin-film hydration technique, and their inhibitory effect was investigated against E. coli O157:H7. The encapsulated essential oils were more effective in diminishing bacterial population compared with their un-encapsulated counterparts. Moreover, nanoliposomal formulations displayed a higher ability to decrease Shiga toxin II gene expression and were able to efficiently control the production of toxins [108].

### 6.5. Flavor and Aroma Encapsulation

The entrapment of flavors is a major challenging area of research in the application of encapsulation technologies in food systems. At present, this application is limited by cost, although the relatively high cost of encapsulation procedures is offset somewhat by a reduction in the quantity of active ingredients required [109]. Costs should decrease as the technology becomes more widespread and improves in efficiency. It should also be possible for the food manufacturer to demand a premium if products with entrapped flavor offer significant advantages to the consumer. In addition to cost effectiveness issues, the delivery of flavoring agents in food systems poses several challenges regarding the preservation and persistence of the highly volatile odorous molecules during manufacturing, storage, packing, and controlled release until they reach the consumers [110,111]. Aroma molecules are generally constituted by alcohols, aldehydes, ketones, esters, or acids of small molecular weight (100–250 Da) [112].

Nanoliposome and tocosome-encapsulated water-soluble aroma molecules and flavoring agents remain entrapped in the aqueous phase of food systems prior to consumption. However, most other encapsulation techniques involve entities that dissolve in water. Tocosomes and nanoliposomes also allow oil-soluble aromas and flavors to be suspended in the aqueous media. Manipulation of the bilayer composition of the vesicles to adjust the phase transition temperature (Tc) allows flavors and aromas to remain entrapped and protected against degradation during storage, but to be released in the mouth. Alternatively, the consumer can instigate vesicle rupture and flavor release immediately before consumption by re-heating the product [72].

Van Nieuwenhuyzen and Szuhaj [113] commented that volatile components are often lost when food is microwaved, and that the presence of lipidic structures, in the form of phospholipid bilayers, helps produce a more desirable flavor-release profile in fat-reduced foods. The entrapment of volatile flavors in the phospholipid vesicles thus has potential in low-fat and/or microwavable food products. The release of these molecules must occur under very specific conditions to stimulate palatability and increase consumer satisfaction. Triggered generally during the heating of the food, the release of aromas and flavors must last during the ingestion time. Modulation of the release kinetics can be achieved by fine tuning the phospho/lipid composition of the carrier systems [114,115].

### 6.6. Encapsulation of Essential Fatty Acids and Essential Oils

Dietary fats are necessary nutrients to ensure a normal metabolism and as a result a good health. These fats play important roles for the transportation, absorption and function of the fat-soluble vitamins (A, D, E, and K), production of cellular ingredients, hormones, and other compounds that are necessary for the proper functioning of the body [116]. The reason that makes the essential fatty acids the most important fats in our diet is their vital role and highly important functions in Human body [117]. Essential fatty acids include polyunsaturated fatty acids (PUFAs) that can vary in the number of carbons in their molecular structure and the presence of carbon-to-carbon double bounds (two or more double bonds). Essential fatty acids are also essential part of cell membranes and influence membrane fluidity and consequently affect the behavior of membrane-bound enzymes and receptors [116,117].

Since Human body is unable to synthesize the essential fatty acids they must be taken by diet [118]. The reason that Human body cannot synthesize omega-3 and omega-6 fatty acids is metabolism deficiency due to which it is unable to add a double bond into a fatty acid with more than nine carbons away from the delta end. These fatty acids are classified in at least four independent families of PUFAs including omega-3, omega-6, omega-7, and omega-9. The most important omega-9 fatty acid is oleic acid, which can be found in almost all fatty foods, however, olive oil is the richest source of this fatty acid. Linoleic and arachidonic acids are the major groups of omega-6 fatty acids and their high content in diet makes them the most notable PUFAs in human body. Cereals, eggs, poultry, majority of vegetable oils, whole grain breads, baked goods and margarine, sunflower, and corn oils are rich in linoleic acid [119]. The main groups of omega-3 fatty acids are alpha-linolenic acid (ALA), eicosapentaenoic acid (EPA) and docosahexaenoic acid (DHA). ALA can be found in canola oil, flaxseed oil, linseed and rapeseed oils, walnuts, and leafy green vegetables. A major source for EPA and DHA is deep cold-water fish and it is believed that they contribute in reducing the risk of congenital heart disease, hypertension, metabolic syndrome X, obesity, atherosclerosis, collagen vascular diseases and cancer [120]. Denaturation and conversion of these fatty acids into trans fats during the processing may turn them into unfavorable products that can be dangerous for health [121,122]. Due to their highly unsaturated structures, these fatty acids are exceptionally susceptible to autoxidation and form complex mixtures of high-molecular-weight polymeric products whose health effects are still unknown [98,123]. Bioactive encapsulation techniques have been successfully employed to stabilize sensitive oils and molecules such as omega-3 for pharmaceutical and food applications. The bilayer membrane of the vesicles improves the shelf life of the encapsulated material by protecting them from undesirable effects such as light, moisture, oxygen and extreme pH values [6,124]. Spray drying, freeze-drying, and coacervation are the most common techniques for the encapsulation of fish oil and essential fatty acids [125]. However, nanoliposomal encapsulation of essential fatty acids and fish oils has also been carried out to improve their stability against oxidation reactions and also to improve their bioavailability [124,125,126].

Ojagh and Hasani [126] incorporated omega-3 fatty acids into nanoliposomes and evaluated the properties of the fortified breads. The encapsulation efficiency of the fish oil was around 90%. Oxidative stability of the encapsulated fish oil was higher than that of free oil. Loaf volume of the breads fortified with nanoliposome was larger, and addition of loaded nanoliposome showed no negative effect on textural quality and sensory properties of the bread. Authors suggested using nanoliposome-encapsulated fish oil in breads and other bakery products to improve their nutritional value [126].

Rasti and co-workers [127] formulated DHA- and EPA-encapsulated nanoliposomes by Mozafari method (as an improved version of heating technique, which allows one-step preparation of nanoliposomes without using toxic solvents or detergents) and evaluated its applicability, stability, and sensory effects in bread and milk in comparison with unencapsulated form of omega PUFAs. The quantity of losses in bread and milk containing omega-encapsulated nanoliposomes was 4.2%–6.5% and 5.6%–5.9%, respectively. However, additional losses did not occur in bread and milk during storage. Sensory evaluation showed no significant difference between the control and loaded nanoliposomes, while the fishy aroma and flavor was detectable in samples containing unencapsulated PUFAs. Milk and breads enriched with omega-3 nanoliposomes had the highest stability during storage while the panelists could not detect the unpleasant fishy flavor. The recovery of omega-3 in samples treated with nanoliposomes was higher compared with other samples and also lower peroxide and anisidine value in nanoliposomal samples were observed. The authors confirmed this method as an effective and reproducible technique for application of omega-3 PUFAs in food systems [127].

Ghorbanzade and colleagues [128] reported similar sensory characteristics for yogurts fortified with nanoliposomal fish oil in comparison with control samples. Higher levels for DHA and EPA after 3 weeks of storage was observed for yogurts fortified with nanoliposomal fish oil than yoghurts containing free fish oil [128]. Rasti and team [129] studied the effect of several factors including mixing rate, mixing time and temperature on the application of Mozafari method in order to prepare nanoliposomes with maximum encapsulation efficiency and the smallest particle size. It was revealed that increase of mixing time generally increased the encapsulation efficiency of nanoliposomes. However, the encapsulation efficiency increased up to a certain level of mixing rate and at the higher mixing rates the efficiency decreased. Increase in all three variables reduced the size of nanoliposomes and by mixing for 60 min at 795 rpm an encapsulation efficiency of about 100% and a mean particle size of 81.4 nm was achieved [129].

Sahari and coworkers [130] developed nanoliposomes containing DHA and EPA with or without alpha-tocopherol. They observed that the particle size of nanoliposomes was statistically stable especially in lower temperatures during storage for 3 months. Lipid oxidation during storage elevated by temperature, and the highest oxidation rate observed was at 40 °C. Although, a high level of EPA and DHA was degraded during storage of nanoliposomes, but the addition of alpha-tocopherol limited EPA and DHA decomposition due to its antioxidant effect, which was more significant for samples stored at 4 °C. These researchers concluded that dual encapsulation of DHA/EPA into antioxidant-containing nanoliposomes can significantly improve the health benefits of food products [130].

In another study, comparison between some physical properties of EPA and DHA incorporated into nanoliposomes prepared by different methods (e.g., extrusion, ultrasonic irradiation, bath sonication, probe sonication, and combined probe and bath sonication) was carried out [131]. The study demonstrated that the most efficient vesicles with the smallest particle size were nanoliposomes prepared by probe and combined probe with bath sonication methods. Furthermore, the highest amount of secondary oxidation products such as propanal, pentanal, hexanal, and heptanal were formed in probe- and bath-sonicated samples. Probe sonication method exhibited the highest loading of DHA and EPA, with a low oxidation rate, and it was suggested for future studies [131].

It has been reported that nanoliposomes prepared using two different types of phospholipids (1,2-dipalmitoyl-3-sn-phosphatidylcholine and 1,2-dioleoyl-3-sn-phosphatidylcholine) were effective to protect DHA from oxidation during lipid peroxidation test [132].

Rasti and colleagues [133] studied the oxidation of PUFAs (DHA and EPA) incorporated into nanoliposomes during storage at 4 °C and compared with bulk PUFAs. Nanoliposomes were prepared by conventional (thin-film hydration using organic solvents) and Mozafari method (a green technology procedure which does not require utilization of toxic organic solvents). PUFA-loaded nanovesicles produced by Mozafari method exhibited the highest physicochemical stability, followed by conventional nanoliposomes and bulk PUFAs. Authors concluded that higher oxidative stability of PUFAs encapsulated by Mozafari method can be attributed to the defensive system from the aqueous phase, which indicates the advantages of using safe solvents over organic solvents during vesicle manufacture [133].

A nutritionally valuable component of certain plant extracts is thymoquinone (TQ). This essential oil exhibits strong antioxidant, anti-inflammatory and antineoplastic effects, both in vitro and in vivo [30]. It is a phytochemical compound found in certain plants including Nigella sativa and black cumin. Successful nanoliposomal formulations of TQ have been reported exhibiting ideal particle characterization (with respect of size and zeta potential) as well as high loading capacity (i.e., around 75%) [31]. More recently, Yaghmur and co-workers formulated nanodispersions composed from binary mixtures of glycerol monooleate and vitamin E stabilized with d-α-tocopheryl poly (ethylene glycol) succinate (TPGS-PEG_2000_) for the encapsulation of thymoquinone [134,135]. The produced nanoparticles were in the form of micellar cubosomes with an internal inverse cubic Fd3m phase and emulsified microemulsions (EMEs) with an internal inverse microemulsion. Authors reported the presence of “flower-like” vesicular populations in both native and drug-loaded nanodispersions. They concluded that the nanodispersions have the potential to accommodate thymoquinone and can be considered as promising formulations for the development of thymoquinone nanomedicines [134,135].

## 7. Summary and Perspectives

Among the myriad of bioactive encapsulation systems, nanoliposomes and tocosomes are the most promising carriers for the entrapment, preservation, and controlled release of hydrophilic, hydrophobic, and amphiphilic compounds owning to their amphipathic nature along with their biocompatibility and safety. Tocosomes and nanoliposomes protect bioactive compounds from degradation within food matrices and inside the GI tract; provide targetability; and improve their solubility, bioactivity, and bioavailability. They minimize the unwanted effects of bioactives on sensorial characteristics and integrity of food products. Furthermore, these nanodelivery systems can be derived from natural and inexpensive sources of ingredient molecules including soy, sunflower, milk, or egg yolk, which make them suitable for large-scale industrial applications. Despite the advantages of these carriers, the main challenges facing the commercial use of nanoliposomes and tocosomes are their low physical stability, relatively high sensitivity to temperature and pH variations, and untimely release of hydrophilic bioactive compounds under long-term storage conditions. Consequently, many investigations have been carried out on the use of compounds that stabilize and modify vesicle structures and also improve the production and storage methods aiming at minimizing disadvantages. Some of the methods for improving the stability and structural modification of nanoliposomes and tocosomes are surface coating of vesicles with biopolymers, application of anionic or cationic components and changes in the composition of phospholipids, application of membrane stabilizing agents such as cholesterol, alpha-tocopherol or glycerol, storage of the vesicles at low temperatures, and lyophilization. However, more studies need to be carried out on the efficiency of the carrier systems under in vivo conditions, optimization of vesicle characteristics, their stability, their effects on different body organs, and their commercialization. Moreover, it is necessary to prepare nanoliposomes and tocosomes without usage of toxic solvents by economical and scalable techniques for pharmaceutical and food sectors.

## Figures and Tables

**Figure 1 molecules-25-00638-f001:**
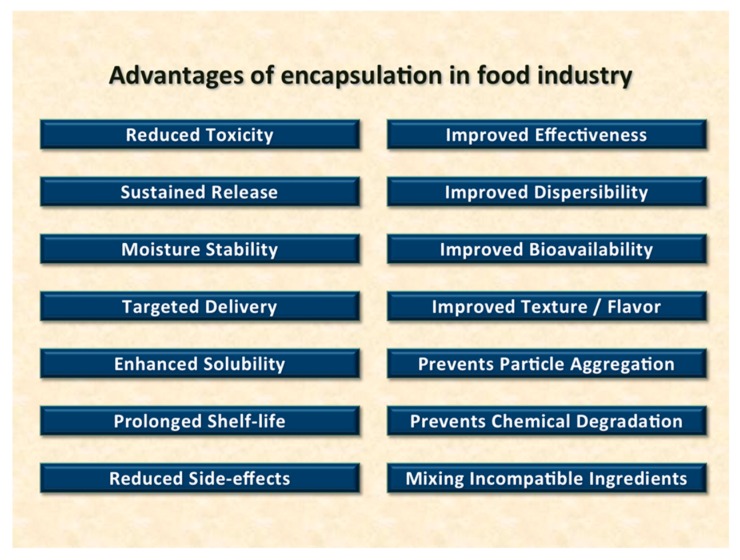
Advantages and benefits of using encapsulation in the food and nutraceutical industries.

**Figure 2 molecules-25-00638-f002:**
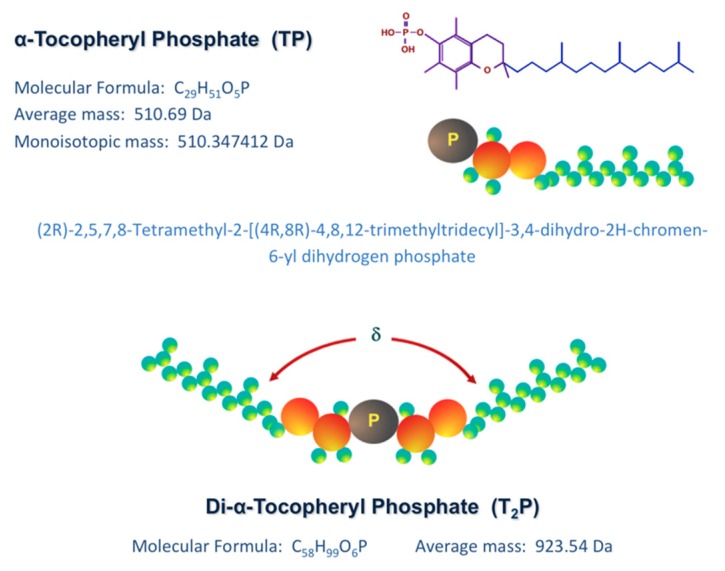
Chemical structure of alpha tocopheryl phosphate (TP) and di-alpha tocopheryl phosphate (T_2_P). The angle of alignment of hydrocarbon chains (phytyl tails) of T_2_P molecule (δ) results in an “inverted truncated cone” molecule with a critical packing parameter (cpp) of greater than 1.

**Figure 3 molecules-25-00638-f003:**
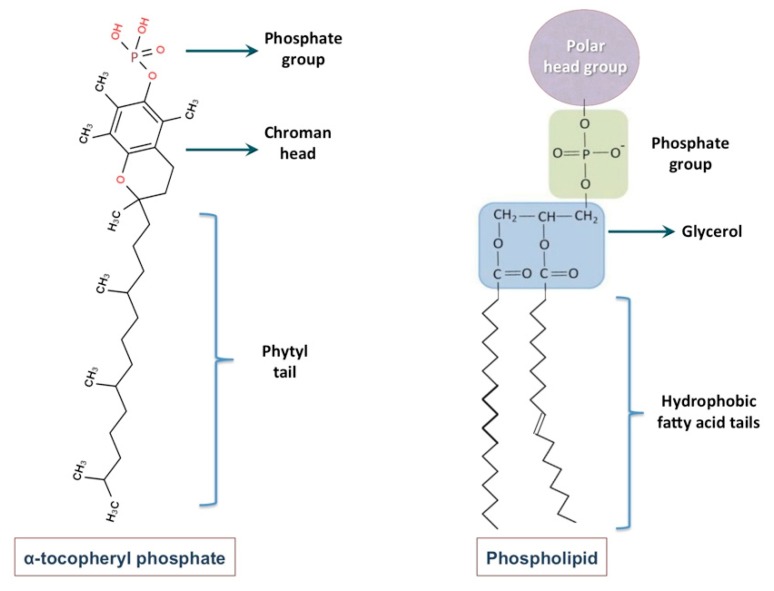
Comparison of the chemical structure of a tocopheryl phosphate molecule with a phospholipid molecule. A phospholipid molecule is composed a phosphate group, 2 fatty acid tails and a glycerol linker. However, alpha-tocopheryl phosphate consists of a chroman head (with two rings: one phenolic and one heterocyclic) and a phytyl tail with 3 isoprene side-chains.

**Figure 4 molecules-25-00638-f004:**
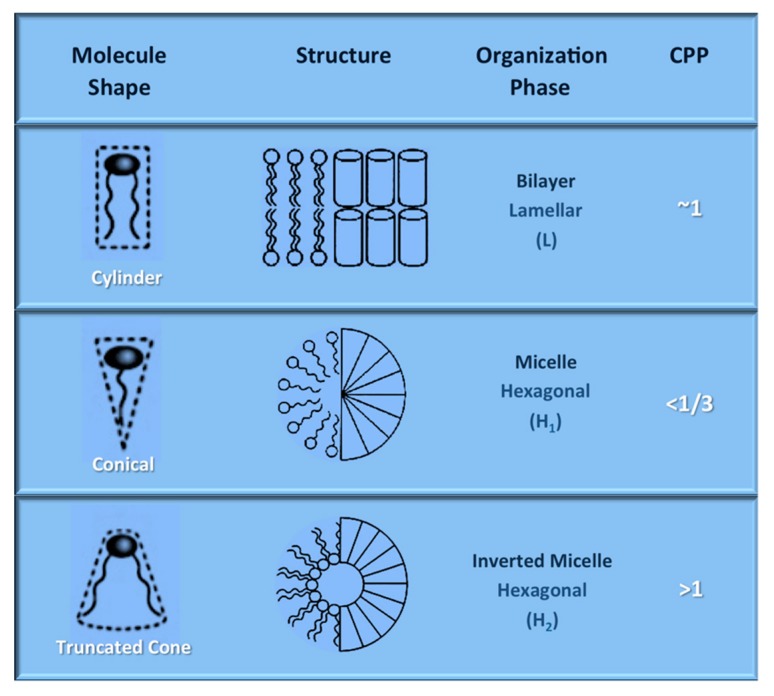
Schematic diagram of the effect of molecular shape on the structure of the three main types of amphiphilic aggregates. CPP: critical packing parameter (adapted with modifications from Kulkarni 2016; Lasic 1998).

**Figure 5 molecules-25-00638-f005:**
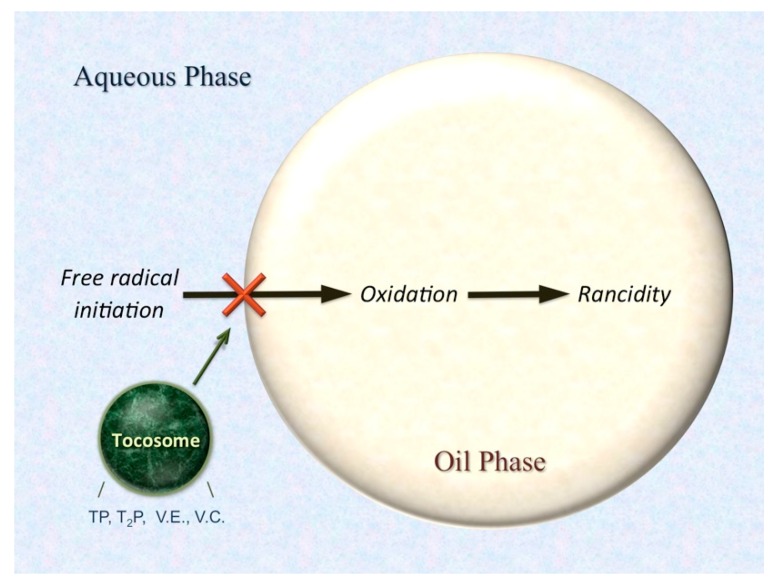
Protection of a food emulsion by multi-antioxidants entrapped in a tocosome. TP: tocopheryl phosphate; T_2_P: di-tocopheryl phosphate; VE: vitamin E (alpha tocopherol); VC: vitamin C (ascorbic acid and ascorbyl palmitate).

**Table 1 molecules-25-00638-t001:** Some of the commonly used preparation methods of bioactive carriers (including nanoliposomes and tocosomes). From References [2,8,20,33,38,39].

Method	Advantages	Disadvantages
Thin-film hydration method	High solubility of ingredients in the initial stage of the process	Use of potentially toxic solvents, time consuming, difficult to scale-up
Ethanol/ether injection	Simple procedure	Organic solvent residue, nozzle blockage in ether system, time consuming, sterilization issue
Reverse phase evaporation	Simple design, acceptable encapsulation efficiency	Not suitable for the encapsulation of sensitive material due to large quantity of organic solvent use, time consuming, sterilization issue
Microfluidisation	Control of particle size, large volume manufacture in a continuous and reproducible manner	Employment of high pressures (up to l0,000 psi)
Supercritical Fluid Process (SFP)	Control of particle size, possibility of in situ sterilization, low organic solvent consumption	High cost, low yield, high pressure up to 350 bar used
Dual asymmetric centrifugation	Simple method, yields products with narrow size distribution, high encapsulation efficiency	Not suitable for bulk production, high pressure and high shear force
Sonication	Simple and fast technique	Overheating of the sample causing degradation, sonicator tips releases metal particles into the product
Heating Method	Organic solvent free, scalable	High temperature requirement
Mozafari Method	Simple design, safe and mild procedure, organic solvent free, easily scalable	New method, Reproducibility need to be attested under different conditions
Binary Nanodispersions	Organic solvent free, not requiring secondary emulsifier	Requires ultrasonication

**Table 2 molecules-25-00638-t002:** Some reported studies on the nanoencapsulation of natural food preservatives and antimicrobial agents. From References [41,102,103,104,105,106].

Formulation/Method	Targeted Microorganisms	Encapsulated Antimicrobial
Dynamic high-pressure microfluidization	*L. monocytogenes*, *S. aureus*	Eugenol
Ethanol injection	*Escherichia coli*	Garlic essential oil
Freeze-thaw method	*Microsporum gypseum*, *Microsporum canis*, *Trichophyton verrucosum*, *Trichophyton rubrum*	Essential oil of Eucalyptus camaldulensis leaf
Mozafari method	*Bacillus subtilis*, *Pseudomonas aeruginosa*	Nisin
Proliposome	*L. monocytogenes*	Nisin
Sonication	*L. monocytogenes*, *E. coli*, *Penicillium chrysogenum*, *Botrytis cinerea*	d-limonene
Thin-film hydration method	*S. aureus*, *E. coli*	Clove oil

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
