# Peer review of "Nanoliposomes and Tocosomes as Multifunctional Nanocarriers for the Encapsulation of Nutraceutical and Dietary Molecules"

_molecules, 2020, doi:10.3390/molecules25030638_

Round 1

Reviewer 1 Report

This is a review manuscript on the potential use of liposomes and tocosomes as nanocarriers for the delivery of nutraceuticals. The manuscript is suitable for publication in Molecules after a major revision. The main areas for improving this manuscript that I could identify were in (i) reducing the verbosity in some areas of the manuscript, (ii) improving the text by avoiding repetitions and inserting relevant references, and (iii) improving the clarity of different sections.

Main and Minor technical comments:

The authors should avoid repeating same information in the text. For instance, repetitions related to the advantages of liposomes and tocosomes. I would recommend using either ‘liposomes or ‘nanoliposomes’. It is worth avoiding using both of them in the text. The attractive liposomal formulations have nanoscale dimensions so it is confusing when using both terms in the text. Abstract and other sections in the manuscript: it is worth describing in more precise manner liposomes and other relevant lipid nanoparticles. In this respect, I would recommend adding the word ‘generally’ before ‘spherical’ in line 23, page 1. The authors should also consider that not all liposomes can be formed by applying low- or high-energy input. It is worth mentioning in the introduction recent studies on the potential use of other lipid nanoparticles relevant to liposomes and tocosomes such as cubosomes and hexosomes in the development of nanocarriers for the delivery of nutraceuticals. See for example: Molecules 2020, 25(1), 16; the book chapter: “Nanoencapsulation of Food Ingredients by Cubosomes and Hexosomes“, In: “Lipid-Based Nanostructures for Food Encapsulation Purposes”, S. M. Jafari (ed.) It is important to add relevant references after different sentences written in the manuscript. Line 25, page 1: to replace ‘structure’ by ‘objects’. In Line 26: ‘biphasic’ can be replaced by ‘bi-compartmental’ Line 27, page 1 and in different sentences: it is worth replacing ‘controlled manner’ by ‘sustained manner’ Line 30, page 1: to replace ‘material’ by ‘materials’. Overall: the authors should check carefully the text Line 36, page 1: to replace ‘tocosome technology’ by ‘tocosomes’. For the first paragraph of introduction: I would avoid writing ‘encapsulation and / or entrapment’. It is also not clear what the authors mean by vesicular system or coating. I would recommend focusing on encapsulation of nutraceuticals. Lines 63 & 64: replacing ‘lipid and/or phospholipid molecules’ by ‘phospholipids’ In figure 1: is it worth mentioning ‘reduced toxicity’? Reducing toxicity is considered an advantage when encapsulating drugs in liposomes and other similar lipid nanoparticles. Nutraceuticals are food-grade and safe. This is not clear for me Line 74, page 2: to replace ‘become’ by ‘becomes’. The authors should read carefully the text. Line 76, page 2: to replace ‘lipidic phase’ by ‘lipidic domains’. In addition, there still need for several changes in the text. The authors should read carefully the written text. For example, the word ‘feed’ should be deleted (page 3, line 83). For table 1: the authors could consider the following recent work on the preparation of liposomes and other lipid nanoparticles: J. Control. Rel. 2016, 239, 1–9.

Author Response

REVIEWER 1:

The authors should avoid repeating same information in the text. For instance, repetitions related to the advantages of liposomes and tocosomes.

In accordance with the suggestion of Reviewer repetitive sentences have been modified. As a result, the Manuscript covers different advantages of liposomes and tocosomes in a non-repetitive way and these are briefly mentioned in the final section (7. Summary and Perspectives).

I would recommend using either ‘liposomes or ‘nanoliposomes’. It is worth avoiding using both of them in the text. The attractive liposomal formulations have nanoscale dimensions so it is confusing when using both terms in the text.

Manuscript has been amended with respect of proper use of terminologies "liposomes" and "nanoliposomes" as recommended. Use of the mentioned terminologies is based on the definition, and associated Reference, given on Page 2 of the manuscript: "The nano-scale form of liposomes are called nanoliposomes [8]." This definition is highlighted in the manuscript.

Abstract and other sections in the manuscript: it is worth describing in more precise manner liposomes and other relevant lipid nanoparticles. In this respect, I would recommend adding the word ‘generally’ before ‘spherical’ in line 23, page 1. The authors should also consider that not all liposomes can be formed by applying low- or high-energy input.

We thank the Reviewer for technical suggestions. The word "generally" is now added before "spherical" in the Abstract (Line 23) as well as in Line 229 under Section 5.

It is worth mentioning in the introduction recent studies on the potential use of other lipid nanoparticles relevant to liposomes and tocosomes such as cubosomes and hexosomes in the development of nanocarriers for the delivery of nutraceuticals. See for example: Molecules 2020, 25(1), 16; the book chapter: “Nanoencapsulation of Food Ingredients by Cubosomes and Hexosomes“, In: “Lipid-Based Nanostructures for Food Encapsulation Purposes”, S. M. Jafari (ed.).

Suggested publication of Dr Jafari, along with his other recent publications, have already been utilized in different sections of the manuscript including Introduction. A new paragraph has been added at the end of section 6.6. to accommodate highly interesting study by Dr Yaghmur et al. (Molecules 2020, 25(1), 16).

It is important to add relevant references after different sentences written in the manuscript.

We believe adequate number of References have been cited by the Manuscript (154 References in total).

Line 25, page 1: to replace ‘structure’ by ‘objects’. In Line 26: ‘biphasic’ can be replaced by ‘bi-compartmental’.

Suggested corrections are done.

Line 27, page 1 and in different sentences: it is worth replacing ‘controlled manner’ by ‘sustained manner’.

Corrections done as suggested.

Line 30, page 1: to replace ‘material’ by ‘materials’. Overall: the authors should check carefully the text.

Corrections are done and the manuscript text has been checked as suggested.

Line 36, page 1: to replace ‘tocosome technology’ by ‘tocosomes’.

The phrase 'tocosome technology' is replaced by 'tocosomes'

For the first paragraph of introduction: I would avoid writing ‘encapsulation and / or entrapment’. It is also not clear what the authors mean by vesicular system or coating. I would recommend focusing on encapsulation of nutraceuticals.

By 'encapsulation' we specifically refer to the accommodation of active material inside the aqueous core of the vesicles. The word 'entrapment' is a broad terminology and can refer to the presence of the active material inside the bilayer (lipid phase) of the nanoliposomes or tocosomes, its internal aqueous core or electrostically attached to the external surface of the vesicles. Therefore, we would retain the present wording of the text.

The word 'coating' is deleted in the mentioned section.

Lines 63 & 64: replacing ‘lipid and/or phospholipid molecules’ by ‘phospholipids’.

Correction is done as suggested.

In figure 1: is it worth mentioning ‘reduced toxicity’? Reducing toxicity is considered an advantage when encapsulating drugs in liposomes and other similar lipid nanoparticles. Nutraceuticals are food-grade and safe. This is not clear for me

Certain food and nutraceutical ingredients are toxic at high concentrations (authors are lecturers of the course 'Food Toxicology'). When encapsulated, by controlling the released dose, this toxicity concern will be rectified.

Line 74, page 2: to replace ‘become’ by ‘becomes’. The authors should read carefully the text.

Correction is done as suggested.

Line 76, page 2: to replace ‘lipidic phase’ by ‘lipidic domains’.

Correction is done as suggested.

In addition, there still need for several changes in the text. The authors should read carefully the written text. For example, the word ‘feed’ should be deleted (page 3, line 83).

The phrase 'feed' refers to "animal food". The phrase and sentence is scientifically correct.

For table 1: the authors could consider the following recent work on the preparation of liposomes and other lipid nanoparticles: J. Control. Rel. 2016, 239, 1–9.

We would like to thank this Reviewer for valuable suggestions towards improving the manuscript. The suggested Reference is added to Table 1.

__________________________________

Reviewer 2 Report

This review on the use of nanaocarriers in some specific defined area is well documented. The use of the term liposomes and nanoliposomes is somehow misleading and the authors should clearly define in the introduction the differences between both. It's not clear what are the difference for a non specialists. How they compare nanoliposomes to LUV, or better SUV ?

In table 1 the classical freeze/thawing and extrusion methods are missing, is there a reason ?

Also they authors could/should add some comments of the uses of cubosomes in diary food industry as this kind of carriers have also been extensively studied in the past.

They should also add some structural parameters of the different carrier, typical size, zeta potential, stability etc.

The authors could also point better the general interest of the nanoliposome approach.

Author Response

REVIEWER 2:

This review on the use of nanaocarriers in some specific defined area is well documented. The use of the term liposomes and nanoliposomes is somehow misleading and the authors should clearly define in the introduction the differences between both. It's not clear what are the difference for a non specialists. How they compare nanoliposomes to LUV, or better SUV ?

We thank the Reviewer for expert comments and suggestions.

The word "liposome" is deleted in several places in the manuscript.

Scientific definition of 'liposomes' and 'nanoliposomes' is given on Page 2. Details of differences between different types of liposomes and nanoliposomes, including SUV, MLV and LUV, are detailed in our previous publications.

In table 1 the classical freeze/thawing and extrusion methods are missing, is there a reason ?

Typical methods pertaining to the field of food and nutraceuticals are listed in the Table. The extrusion method is used as a down-sizing procedure along with some of the listed methods. Recent References provided in the Table and text cover more exhaustive list of the nanoliposome and tocosome preparation methods.

Also they authors could/should add some comments of the uses of cubosomes in diary food industry as this kind of carriers have also been extensively studied in the past.

A new paragraph has been added at the end of section 6.6. which includes cubosome application in food industry.

They should also add some structural parameters of the different carrier, typical size, zeta potential, stability etc.

Results of discussed References pertaining to the mentioned parameters are given in the manuscript. In the added new paragraph (at the end of section 6.6.) more examples of the mentioned parameters are provided.

The authors could also point better the general interest of the nanoliposome approach.

New References on the topic have been added (including one 2020 citation) to emphasize the importance of nanoliposomes in the field.

__________________________________

Reviewer 3 Report

The paper of Mozafari et al. reviews the applications of liposomes (why nanoliposomes?, the unique liposomes not nano are giant liposomes!!!) and the so-called "tocosomes" in Food Sciences.

Some of the applications indicated were reported 30-35 years ago, and one can find them in several old reviews. This review is most suitable for a journal dealing with dairy products of food science and technology than for Molecules. The reader of Molecules seeks other kind of information.

The references indicated are not the seminal works in the field of liposomists. Contrarily, an excessive number of autocites are in the references listed.

Author Response

REVIEWER 3:

The paper of Mozafari et al. reviews the applications of liposomes (why nanoliposomes?, the unique liposomes not nano are giant liposomes!!!) and the so-called "tocosomes" in Food Sciences.

We would like to thank the Reviewer for useful comments and suggestions.

The word "liposome" is deleted in several places in the manuscript as suggested.

Scientific definition of 'liposomes' and 'nanoliposomes' is given on Page 2.  

Some of the applications indicated were reported 30-35 years ago, and one can find them in several old reviews. This review is most suitable for a journal dealing with dairy products of food science and technology than for Molecules. The reader of Molecules seeks other kind of information.

Out of 154 References, 58 are published since 2017. We strongly believe the information are completely new. nanoliposomes are relatively new encapsulation platform and Tocosomes have been just introduced in 2017.

The references indicated are not the seminal works in the field of liposomists. Contrarily, an excessive number of autocites are in the references listed.

The authors being inventors of 'Tocosome' and 'Mozafari Method' had to use some autocitations. We are also co-authors of the first book on 'Nanoliposomes'.

___________________________

Round 2

Reviewer 1 Report

The manuscript is certainly improved. I have only the following minor technical comments:

line 25, abstract: to replace 'object' by 'objects'. There is misunderstanding regarding the references. My comment on the manuscript was not related to the number of references. I just asked to insert relevant references after different sentences in the text. For instance, it is worth adding relevant reference(s) after the first sentence in the introduction, first sentence of the second paragraph of the introduction. There is also need to consider that in different sections. Line 718: to replace 'The  nanostructures  were  in  the  form  of  cubosomes  and  inverse  micellar...' by 'The produced nanoparticles were in the form of micellar cubosomes with an internal inverse cubic Fd3m phase and emulsified microemulsions (EMEs) with an internal inverse microemulsion'.  

Author Response

Reviewer comments:

Comments and Suggestions for Authors
The manuscript is certainly improved. I have only the following minor technical comments:

line 25, abstract: to replace 'object' by 'objects'. There is misunderstanding regarding the references. My comment on the manuscript was not related to the number of references. I just asked to insert relevant references after different sentences in the text. For instance, it is worth adding relevant reference(s) after the first sentence in the introduction, first sentence of the second paragraph of the introduction. There is also need to consider that in different sections. Line 718: to replace 'The nanostructures were in the form of cubosomes and inverse micellar...' by 'The produced nanoparticles were in the form of micellar cubosomes with an internal inverse cubic Fd3m phase and emulsified microemulsions (EMEs) with an internal inverse microemulsion'.

Our response:

We have complied with all suggestions of the reviewer and have done the changes accordingly.  Changes to the manuscript are highlighted in blue color. 
